# AriEL: VOLUME CODING
# FOR SENTENCE GENERATION COMPARISONS

## ABSTRACT

Mapping sequences of discrete data to a point in a continuous space makes it difficult to retrieve those sequences via random sampling. Mapping the input to a volume would make it easier to retrieve at test time, and that is the strategy followed by the family of approaches based on Variational Autoencoder. However the fact that they are at the same time optimizing for prediction and for smoothness of representation, forces them to trade-off between the two. We benchmark the performance of some of the standard methods in deep learning to generate sentences by uniformly sampling a continuous space. We do it by proposing AriEL, that constructs volumes in a continuous space, without the need of encouraging the creation of volumes through the loss function. We first benchmark on a toy grammar, that allows to automatically evaluate the language learned and generated by the models. Then, we benchmark on a real dataset of human dialogues. Our results indicate that the random access to the stored information can be significantly improved, since our method AriEL is able to generate a wider variety of correct language by randomly sampling the latent space. VAE follows in performance for the toy dataset while, AE and Transformer follow for the real dataset. This partially supports the hypothesis that encoding information into volumes instead of into points, leads to improved retrieval of learned information with random sampling. We hope this analysis can clarify directions to lead to better generators.

## 1 INTRODUCTION

It is standard for neural networks to map an input to a point in a $d$-dimensional real space (Hochreiter and Schmidhuber, 1997; Vaswani et al., 2017; LeCun et al., 1989). However, that makes it difficult to find a specific point when the real space is being sampled randomly. That can limit the applicability of pre-trained models to their initial scope. Some approaches do map an input into volumes in the latent space. The family of approaches that stems out of the idea of Variational Autoencoders (Kingma and Welling, 2014; Bowman et al., 2016; Rezende and Mohamed, 2015; Chen et al., 2018) are trained to encourage such type of representations. By encoding an input into a probability distribution that is sampled before decoding, several neighbouring points in $\mathbb{R}^d$ can end up representing the same input. However, it often implies having two summands in the loss, a log-prior term and a log-likelihood term (Kingma and Welling, 2014; Bowman et al., 2016), that fight for two different causes. In fact, if we want a smooth and volumetric representation, encouraged by the log-prior, it might come at the cost of having worse reconstruction or classification, encouraged by the log-likelihood. Therefore, each diminishes the strength and influence of the other.

By giving partially up on the smoothness of the representation, we propose instead a method to explicitly construct volumes, without a loss that is implicitly encouraging such behavior. We propose AriEL, a method to map sentences to volumes in $\mathbb{R}^d$ for efficient retrieval with either random sampling, or a network that operates in its continuous space. It draws inspiration from arithmetic coding (AC) (Elias and Abramson, 1963) and k-d trees (KdT) (Bentley, 1975), and we name it after them *Arithmetic coding and k-d trEes for Language* (AriEL). For simplicity we choose to focus on language, even though the technique is applicable for the coding of any variable length sequence of discrete symbols. More precisely, we plan to use AriEL in the context of dialogue systems with the goal to provide a tool to optimize interactive agents. The interaction of AriEL with longer text is left as future work.

In particular, it can be used as an objective benchmark to compare other methods and to understand how their latent space is used. AriEL attempts to fill completely the latent space with the sentences present in the training dataset, using notions from information theory. AriEL uses a language model to split the latent space in volumes guided by the probability assigned to the next symbol in a sentence. For this reason, it can simplify as well the reuse of pretrained language models for new tasks and in larger architectures. In fact, it can provide a training agent with a simpler interface with a language model, e.g. a GPT-2 (Radford et al., 2019), where the agent could choose the optimal dimensionality of the interwface. We prove how such a volume representation eases the retrieval of stored learned patterns and how to use it to set references for other models.

Our contributions are therefore:

- AriEL, a volume coding technique based on arithmetic coding and k-d trees (Section 3.1), to improve the retrieval of learned patterns with random sampling;

- the use of a context-free grammar and a random bias in the dataset (Section 3.3), that allows us to automatically quantify the quality of the generated language;

- the notion that explicit volume coding (Section 2 and 5) can be a useful technique in tasks that involve the generation of sequences of discrete symbols, such as sentences.

## 2    RELATED WORK

**Volume codes:** We define a *volume code* as a pair of functions, an encoder and a decoder functions, where the encoder maps an input $x$ into a set that contains compact and connected sets of $\mathbb{R}^d$ (Munkres, 2018), and the decoder maps every point within that set back to $x$. It is a form of distributed representations (Hinton et al., 1984) since the latter only assumes that the input $x$ will be represented as a point in $\mathbb{R}^d$. We define *point codes* as the distributed representations that are not volume codes. Volume codes differ from coarse coding (Hinton et al., 1984) since in this case the code is represented by a list of zeros and ones that identifies in which overlapping sets $x$ falls into. We call *implicit volume codes*, when the volume code is encouraged through a term in the loss function bengio2013representation. Both generative and discriminative models (Ng and Jordan, 2002; Kingma and Welling, 2014; Jebara, 2012) can learn volume codes this way. We call *explicit volume code*, when the volumes are constructed instead through the operations that define the architecture, and are created independently from any loss and optimizer choice.

**Sentence generation through random sampling:** Generative Adversarial Networks (GAN) (Goodfellow et al., 2014) map random samples to a learned generation through a 2-players game training procedure. They have had trouble for text generation, due to the non differentiability of the $argmax$ at the end of the generator, and given that partially generated sequences are non trivial to score (Yu et al., 2017). Several advances have significantly improved their performance for text generation, such as using the generator as a reinforcement learning agent trained through Policy Gradient (Yu et al., 2017), avoiding a binary classification in favor of a cross-entropy for the discriminator that evaluates each word generated (Xu et al., 2018), or with the Gumbel-Softmax distribution (Kusner and Hernández-Lobato, 2016). Random sampling the latent space is used as well by Variational Autoencoders (VAE) (Kingma and Welling, 2014), to smooth the representation of the learned patterns. Training VAE for text has been shown to be possible with KL annealing and word dropout (Bowman et al., 2016), and made easier with convolutional decoders (Severyn et al., 2017; Yang et al., 2017). Several works explore how VAE and GAN can be combined (Makhzani et al., 2015; Tolstikhin et al., 2017; Mescheder et al., 2017). AriEL can be used as a generator or a discriminator in a GAN, or as an encoder or a decoder in an autoencoder. However it differs from them in the explicit procedure to construct volumes in the latent space that correspond to different inputs. The intention is to fill the entire latent space with the learned patterns, to ease the retrieval by uniform random sampling.

**Arithmetic coding and neural networks:** AC is one of the most efficient lossless data compression techniques (Witten et al., 1987; Elias and Abramson, 1963). AC assigns a sequence to a segment in [0,1] whose length is proportional to its frequency in the dataset. AC is used for neural network compression (Wiedemann et al., 2019) but typically, neural networks are used in AC as the model of the data distribution, to perform prediction based compression (Pasero and Montuori, 2003; Triantafyllidis and Strintzis, 2002; Jiang et al., 1993; Ma et al., 2019; Tatwawadi, 2018). We turn AC

into a compression algorithm in $d$ real numbers, to combine its properties with the properties of high-dimensional spaces, which is the domain of neural networks.

**K-d trees and neural networks:** KdT (Bentley, 1975) is a data structure for storage that can handle different types of queries efficiently. It is typically used as a fast approximation to k-nearest neighbours in low dimensions (Friedman et al., 1977). It gives a binary label to the data with respect to its median. It moves through the k dimensions of the data and repeats the process. Neural networks are typically used in conjuction with KdT to reduce the dimensionality of the search space, for KdT to be able to perform queries efficiently (Woodbridge et al., 2018; Yin et al., 2017; Vasudevan et al., 2009). KdT has been used as well in combination with Delaunay triangulation for function learning, as an alternative to NN with Backpropagation (Gross, 1995). A KdT inspired algorithm is used in (Maillard and Solaiman, 1994) to guide the creation of neurons to grow a neural network. We use KdT to make sure that when we make AC turn multidimensional, it makes use of all the space available.

# 3 METHODOLOGY

## 3.1 ARIEL: VOLUME CODING OF LANGUAGE IN CONTINUOUS SPACES

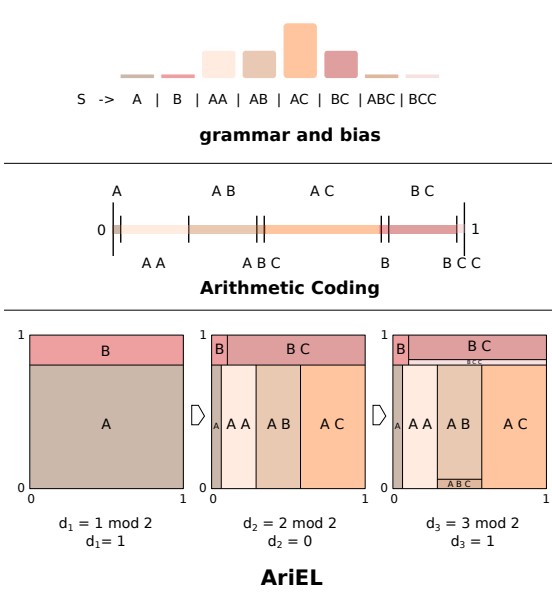

**AriEL**

Figure 1: **Sentence embedding with arithmetic coding and AriEL.** In this example, the generating context-free grammar (CFG) is $S \to A|B|AA|AB|AC|BC|ABC|BCC$, and the bar plot on top indicates the frequency of those sentences in the dataset, as an extra bias to the language. AC (middle) encodes any sequence of this CFG over a single dimension within $[0, 1]$, and the frequency of the sentence determines the length assigned on that segment. AriEL is a multidimensional extension of AC (here in 2D), where the frequency information is preserved in the volumes. The Language Model provides the boundaries where the next symbols are to be found. For a 2D latent space, $d = 2$, the axis to split to find symbol $s_i$ is $d_i = i \mod d$. In the image $d_i = 0$ and $d_i = 1$ represent the horizontal and vertical axis.

AriEL maps the sentence $(s_1, \cdots, s_n)$ to a $d$-dimensional volume of $P((s_1, \cdots, s_n)) = \Pi_{i=1}^n P(s_i|(s_j)_{j<i})$. The sentence is encoded as the center of that volume for simplicity, and any point within it is decoded to the same sentence. Decoding iteratively computes the bounds of the volumes for all possible next symbols and checks inside which bounds the vector is, to find the next symbol at each step. The algorithm is described in detail in the Supplementary Material.

To adapt KdT to more splits than binary, we split the chosen dimension giving a space from 0 to 1 to each possible next symbol, proportional to its probability. The first symbol in the sentence will be assigned a segment of length $P(s_1)$ in the first axis chosen $d_1$, and next symbols will be assigned a segment proportional to their probability conditional to the symbols previously seen e.g. $P(s_3|(s_2, s_1))$ on the axis $d_3$, where $s_1$, $s_2$ and $s_3$ are the first three symbols in the sentence. Then we turn to the following axis and continue the process of splitting and turning (figure 1). We select the next axis in $\mathbb{R}^d$ to split to be $d_i = i \mod d$, where $i \in \{1, 2, \ldots, n\}$ and $n$ is the length of the sequence. If $n$ is larger than the dimension $d$, then the segment in $d_i$ is split again. We applied a neural network to approximate the true statistics of the data $P(s_i|(s_j)_{j<i})$, the Language Model (LM) of AriEL, $P_{LM}(s_i|(s_j)_{j<i})$. This will approximate the frequency information that makes

AC entropically efficient since after a successful training, $P_{LM}(\cdot)$ will converge to $P(\cdot)$. AriEL conserves then the arithmetic coding property of assigning larger volume to frequent sentences.

AriEL only uses a bounded region of $\mathbb{R}^d$, the interval $[0,1]^d$, so encoder and decoder map each input to a compact set and from a compact set. Moreover, they assign sequence $x$ to a hyper-rectangle (Johnson, 2018) and back. Since hyper-rectangles cannot be divided into two disjoint non-empty closed sets, they are connected (Munkres, 2018). Therefore AriEL is a *volume code*. AriEL is an *explicit volume code* since its LM is trained only on a next word prediction log-likelihood loss, without a regularization term that encourages smoothness, and the volumes are constructed by arranging the softmax outputs into a $d$ dimensional grid, operation performed with any choice of loss or optimizer.

AriEL with a RNN-based language model has a computational complexity of $O(nD^2)$ for both encoding and decoding, as it can be seen in algorithms 1 and 2, where $n$ is the length of the sequence and $D$ is the dimensionality of the RNN hidden state. We use capital $D$ to refer to the length of the longest hidden layer among the recurrent layers in the encoder or in the decoder, while $d$ refers to the length of the latent space. AriEL has a minimum number of sequential operations of $O(n)$ for both encoding and decoding, which is on par with conventional recurrent networks for seq2seq learning.

## 3.2 Neural Networks: models and experimental conditions

We compare AriEL to some of the classical approaches to map variable length discrete spaces to fixed length continuous spaces. These are the sequence to sequence recurrent autoencoders (AE) (Sutskever et al., 2014), their variational version (VAE) (Bowman et al., 2016) and Tranformer (Vaswani et al., 2017). We trained them for next word prediction of word $s_i$, when all the previous words are given as input, and they are trained over the biased train set, defined in section 3.3. All of them can be split into an encoder that maps the sentences at the input into $\mathbb{R}^d$, and a decoder that maps back from $\mathbb{R}^d$ into a sentence. More training details can be found in the appendix.

In this work, AriEL's language model neural network $P_{LM}$ consists of a word embedding of size 64, followed by a 140-unit LSTM, a feedforward layer and a softmax over next possible symbols. At test time the argmax is not applied directly to the softmax, but the latent space point is used as the deterministic pointer that chooses the position in cumulative sum of the softmax probabilities. However the language model is trained for next time step prediction through cross-entropy. For both AE and VAE, we stack two GRU layers (Cho et al., 2014) with 128 units at both, the encoder and the decoder, to increase their representational capabilities (Pascanu et al., 2014). Other recurrent networks gave similar results (Hochreiter and Schmidhuber, 1997; Li et al., 2018). The last encoder layer has either $d = 16$ units or $d = 512$ for all methods. The decoder outputs a softmax over the entire vocabulary.

Tranformer (Vaswani et al., 2017) is the state-of-the-art in many S2S problems (Vaswani et al., 2017; Dai et al., 2019; Radford et al., 2018). Since it is a fixed-length representation at the word level but it is variable-length at the sentence level, we padded all sentences to the maximum length in the dataset to be able to compare its latent space capacity to the other models. We take as its latent dimension the connection between the encoder and decoder, with $d_{model}$ size, that will take a value of 16 or 512. We choose most parameters as in the original work (Vaswani et al., 2017): the number of attention heads as $n_{head} = 8$, the key and value dimension as $d_{key} = 64$ and $d_{value} = 64$, a dropout regularization of 0.1, and we only change the stack of identical decoders and encoders to $n_{layers} = 2$, and the dimension of the inner feed-forward network to $d_{inner\ layer} = 256$ to have a number of parameters similar to the other methods. In the GuessWhat?! dataset we tested $n_{layers} = 20$ to have an amount of parameters comparable for $d = 16$ to the other methods, but performed worse than $n_{layers} = 2$, so we report the latter.

## 3.3 Datasets: toy and human sentences

We perform our analysis on two datasets. A toy dataset of sentences generated from a context-free grammar (CFG) and a realistic dataset of sentences written by humans playing a cooperative game.

**The toy dataset:** we generate questions about objects with a CFG (appendix 1). To stress the learning methods and understand their limits we choose a CFG with a large vocabulary and numerous

grammar rules, rather than more classic alternatives (e.g. REBER). The intention is as well to focus on dialogue agents and that's the reason why all sentences are framed as questions about objects.

We distinguish between *unbiased* sentences, those that have been simply sampled from the CFG, and *biased* sentences, those that after being sampled from the CFG have been selected according to an additional structural constraint. To do so we generate an adjacency matrix of words that can occur together in the same sentence, and we use that as the filter to bias the sentences. Once a sentence is produced from the CFG, if all its words can be together in a sentence judged by the adjacency matrix, the sentence is considered as biased, and unbiased otherwise. For simplicity the adjacency matrix is a random matrix of zeros and ones, generated only once for all the experiments, making sure that some symbols such as *the*, *it* or *?*, can be found in both types of sentences. The intention is to emulate the setting were a CFG is constrained by realistic scenes, where not all the grammatically correct sentences are semantically correct: e.g. "Is it the wooden shower in the kitchen ?" could be grammatical, but semantically incorrect given that it is unusual in a realistic scene. We use it to detect how each learning method is able to extract the grammar and extract the roles of each word, despite a bias that makes it harder.

The vocabulary consists of 840 words. The maximal and mean length of the sentences is of 19 and 9.9 symbols. We split the biased dataset into 1M train, 10k test and 512 validation sentences, where no sentence is shared between sets. We created another set of 10k unbiased test sentences with the same CFG, where we only gather sentences that do not follow the adjacency matrix, to make sure that the overlap of this test set is zero with previous ones. We train on the biased sentences and we test if they grasped the grammar behind, with the unbiased.

**The real dataset:** we choose the GuessWhat?! dataset (De Vries et al., 2017), a dataset of sentences asked by humans to humans to solve a cooperative game. It has a vocabulary of 10,469 words, an order of magnitude larger than the toy CFG. The maximal and mean length of the sentences are of 57 and 5.9 symbols.

## 3.4 EVALUATION METRICS

### 3.4.1 QUALITATIVE EVALUATIONS

The two qualitative studies are: (1) we list a few samples of reconstruction via next word prediction of unbiased sentences, to understand the generalization capabilities of the different models (table 1), (2) we list a few samples of generated sentences when the latent space is sampled randomly, to understand the generation capabilities (table 2).

### 3.4.2 QUANTITATIVE EVALUATIONS ON THE TOY GRAMMAR, CFG

We propose measures for 3 properties of an autoencoder: the quality of generation, prediction and generalization. We perform our studies for networks with a latent dimension of 16 units, to understand their compression limits, and for 512 units, which is often taken as the default size (Kingma and Welling, 2014; Vaswani et al., 2017).

**Generation/Decoding Quality** is evaluated with sentences produced by the decoder when the latent space of each model is sampled randomly. The sampling is done uniformly in the continuous latent space, within the maximal hyper-cube defined by the encoded test sentences. We sample 10k sentences and apply four measures: *i) grammar coverage (GC)* as the number of grammar rules (e.g. single adjective, multiple adjectives) that could be parsed in the sampled sentences, over four, the maximal number of adjectives plus one for sentences without adjectives; *ii) vocabulary coverage (VC)* as the ratio between the number of words in the sampled sentences, over 840, the size of the complete vocabulary; *iii) uniqueness (U)* as a ratio of unique sampled sentences; and *iv) validity (V)* as a ratio of valid sampled sentences, sentences that were unique and grammatically correct. We keep our definition of grammar rule simple, for computational efficiency, and for clarity, given that the grammar tree is defined with an ambiguous number of placeholders and terminal symbols.

**Prediction Quality** is evaluated by encoding and decoding the 10k *biased* test sentences as follows: *i) prediction accuracy biased (PAB)* as a ratio of correctly reconstructed sentences (i.e. all words must match); *ii) grammar accuracy (GA)* as a ratio of grammatically correct reconstructions (i.e.

can be parsed by the CFG, even if the reconstruction is not accurate). and *iii) bias accuracy (BA)* as the ratio of inaccurate reconstructions that are still grammatical and keep the bias of the training set.

**Generalization Quality** is evaluated using the 10k *unbiased* test sentences while the embeddings were trained on the *biased* training set. The *prediction accuracy unbiased (PAU)* is computed in the same way as *PAB*, as the ratio of correctly reconstructed ubiased sentences. It allows us to measure how well the latent space generalizes to grammatically correct sentences outside the bias.

### 3.4.3 Quantitative evaluations on the real dataset, GuessWhat?!

In a real dataset we don't have a notion of what is grammatically correct, since humans can use spontaneously ungrammatical constructions. We quantified the quality of the language learned with two measures: *uniqueness* is the percentage of the sentences generated with random sampling that was unique over the 10K generations and *validity* was the percentage of the unique sentences that could be found in the training data, indicating how easy it was to retrieve the learned information.

### 3.4.4 Quantitative evaluations: random interpolations within AriEL

In figure 2 we show what we call the interpolation diversity given the dimension of the latent space. It measures how many of the sentences generated through a straight line between two random points in $\mathbb{R}^d$ were unique and grammatically correct for AriEL for different values of $d$. The Language Model tested is the one trained on the toy grammar.

## 4 Results

### 4.1 Qualitative Evaluations

We present the qualitative studies performed for $d = 16$. Table 1 shows the output of the generalization study. To avoid cherry picking, we display the first 4 reconstructed sentences. AE and VAE fail to generalize to the unbiased language, however both manage to keep the structure at the output of the input sentence. Their behavior improved significantly when the latent space dimension is increased to $d = 512$, with the corresponding increase of parameters. In theory, AriEL is able to reconstruct any sequence by design, by keeping a volume for each of them. However in practice, it failed only slightly less often than the Transformer. Both produce reconstructions of the unbiased input at a similar rate, as it can be see in table 1 and in the metric PAU in table 3 and figure S2. This means that to a reasonable degree, the areas that represent unseen data during training, are available and relatively easy to track for AriEL and Transformer. Instead, all the latent space seems to be taken almost exclusively by the content of the training set for AE and VAE, since sentences that are not seen during training (the unbiased sentences) cannot be reconstructed at all.

The generation study is shown in Table 2 (first 4 samples for each model). AriEL excels at this task, and almost all generations are unique and grammatically correct (*valid*). AE and VAE perform remarkably well given the small latent space. As it is shown in the quantitative study, VAE almost triples AE performance in terms of generation of valid sentences when $d = 16$ (table 3). Transformer performs poorly at generating grammatical sentences when the latent space is sampled randomly. The quantitative analysis reveals however that with the increase of the latent space, Transformer, AE and VAE achieve all improved validity, remaining at one third the reference set by AriEL.

### 4.2 Quantitative Evaluations

The results of the quantitative study are shown in table 3 and in figure S2. AriEL outperforms or closely matches every other method for all the 8 measures, outperforming the rest by a large margin for validity, i.e. unique and grammatical sentences generated, the most important of the metrics. Transformer performs remarkably well at not overfitting and it is able to reconstruct biased and unbiased sentences better than the other non-AriEL methods, even under-parameterized ($d = 16$). It manages to cover all grammar rules in generation but it performs very poorly at generating a diverse set of valid sentences by random sampling. It only needed one iteration through the data to achieve almost perfect validation accuracy, without losing performance when we trained for the remaining 9 epochs. VAE 16 despite the poor generalization to the biased and the unbiased test set,

**Input Sentences**
is the thing this linen carpet made of tile ?
is it huge and teal ?
is the thing transparent , huge and slightly heavy ?
is the object antique white , tiny and closed ?
**AriEL**
is the thing this lime carpet made of tile ?
is it huge and teachable ?
is the thing transparent , huge and slightly heavy ?
is the object antique white , tiny and closed ?
**Transformer**
is the thing this stretchable carpet made of tile ?
is it huge and magenta ?
is the thing transparent , huge and slightly heavy ?
is the object antique white , tiny and closed ?
**AE**
is the thing this small toilet made of laminate ?
is it this average-sized and average-sized laminate ?
is the thing very heavy , heavy and very heavy ?
is the object light pink , small and textured ?
**VAE**
is the thing a small and textured deep stone ?
is it the light deep bedroom ?
is the thing textured , textured and moderately heavy ?
is the thing light , moderately heavy and light green ?

**AriEL**
is the object that tiny very light set ?
is the thing a tiny destroyable abstraction ?
is the thing this mint cream textured organic structure ?
is the object this small large wearable textile ?
**Transformer**
is the thing slightly heavy heavy stone squeezable closed sea heavy ?
is it pale lime executable executable shallow decoration drab turquoise , heavy and potang ?
is it an tomato slot box made of decoration facing stone ?
is the thing short and spring heavy slightly heavy potang ?
**AE**
is the object that light light laminate ?
is the thing a light , small and small laminate ?
is the thing that tiny small decoration stone ?
is the object the average-sized , textured and average-sized laminate ?
**VAE**
is the thing a light and deep office ?
is it light , light and light and pink ?
is the object dark , light and pink ?
is the object a light deep living room ?

Table 1: **Generalization: next word prediction of unbiased sentences at test time.** An unbiased sentence is encoded and decoded by each model. Color means that the word was incorrectly reconstructed. Blue means that the sentence complies with the bias and purple means that the incorrect reconstruction is still unbiased. Most reconstructions seem grammatically correct. In practice AriEL also made errors. Some of its failed reconstructions comply with the training bias, some do not. Transformer performs remarkably well, and interestingly the errors made tend to turn the unbiased input sentence into a biased version at the output. AE produced only biased sentences whose structure resembled the unbiased ones. VAE behaved similarly, producing more unbiased sentences.

Table 2: **Generation: output of the decoder when sampled uniformly in the latent space.** Red defines grammatically incorrect generations according to the CFG the models are trained on. AriEL produces an extremely varied set of grammatically correct sentences, most of which keep the bias of the training set. Transformer reveals itself to be hard to control via random sampling of the latent space, since it almost never produces correct sentences with this method. AE and VAE manage to produce several different sentences, the latter producing more non grammatical, but as well more varied grammatical ones.

results in the best non-AriEL generator, measured by validity. The conflict between log-prior and log-likelihood, encouraged VAE to look for sentences outside the bias, since it was able to produce more grammatically correct sentences, albeit unbiased, than AE. Increasing the learned parameters ($d = 512$), had no effect on Transformer, that was already excellent in several of the metrics, apart from a significant improvement in validity. However, a larger latent space and the increase in number of parameters that followed, prevented AE and VAE from overfitting (better PAU and PAB).

When trained on human sentences, on the GuessWhat?! dataset, AriEL sets again a large validity to be reached. Every approach seems to generate more unique sentences than AriEL, but the fraction of them that is a good generation is very small. Less than $6\%$ of the unique sentences generated by AE, VAE and Transformer are in the training set, while AriEL achieves $22.47\%$ and more.

In the interpolation diversity study (figure 2) we see that for low $d$, we have to pass through many sentences in between two random points in the latent space, while as we augment the dimensionality, we distribute the sentences in different directions. Therefore we find less sentences when we move on the straight line between two random points. The specific curve, lower threshold and speed of decay, will vary for different vocabulary sizes and given the complexity of the language learned.

| | | Generation | | | | Prediction | | | Generalization |
|---|---|---|---|---|---|---|---|---|---|
| | param | grammar coverage | vocabulary coverage | validity | uniqueness | bias accuracy | grammar accuracy | prediction accuracy biased | prediction accuracy unbiased |
| $d = 16$ | | | | | | | | | |
| AriEL | 237K | 100.0 ± 0.0% | 70.4 ± 0.2% | 97.6 ± 0.2% | 99.7 ± 0.1% | 100.0 ± 0.0% | 100.0 ± 0.0% | 100.0 ± 0.0% | 53.1 ± 0.4% |
| Transformer | 258K | 100.0 ± 0.0% | 70.1 ± 0.8% | 4.7 ± 2.7% | 99.1 ± 0.5% | 99.98 ± 0.01% | 99.95 ± 0.02% | 99.92 ± 0.02% | 49.0 ± 0.1% |
| AE | 258K | 100.0 ± 0.0% | 6.89 ± 0.7% | 11.5 ± 4.2% | 13.9 ± 5.1% | 89.5 ± 2.3% | 98.0 ± 1.7% | 0.0 ± 0.1% | 0.0 ± 0.1% |
| VAE | 258K | 100.0 ± 0.0% | 11.5 ± 2.6% | 16.0 ± 9.2% | 24.3 ± 14.8% | 85.4 ± 5.2% | 85.1 ± 8.8% | 0.0 ± 0.1% | 0.0 ± 0.1% |
| $d = 512$ | | | | | | | | | |
| AriEL | 237K | 100.0 ± 0.0% | 70.2 ± 0.3% | 97.9 ± 0.2% | 99.8 ± 0.1% | 100.0 ± 0.0% | 100.0 ± 0.0% | 100.0 ± 0.0% | 53.2 ± 0.3% |
| Transformer | 9M | 100.0 ± 0.0% | 67.3 ± 0.9% | 17.2 ± 6.3% | 87.2 ± 7.5% | 99.99 ± 0.01% | 99.91 ± 0.03% | 99.86 ± 0.05% | 49.0 ± 0.1% |
| AE | 120M | 100.0 ± 0.0% | 39.3 ± 6.0% | 21.0 ± 11.8% | 71.8 ± 5.6% | 82.2 ± 3.5% | 86.8 ± 1.3% | 34.7 ± 11.4% | 24.4 ± 6.0% |
| VAE | 120M | 85.0 ± 12.6% | 28.9 ± 2.4% | 26.5 ± 2.4% | 95.2 ± 3.8% | 73.8 ± 2.2% | 89.5 ± 2.8% | 4.3 ± 3.7% | 4.9 ± 3.6% |

Table 3: **Evaluation of continuous sentence embeddings on the toy dataset.** Results for a latent space of $d = 16$ and $d = 512$. Each experiment is run 5 times. AriEL, achieves almost perfect performance in most metrics, especially in validity, which quantifies how many random samples were decoded into a unique and grammatical sentence. Transformer performed exceptionally, except for validity. All methods improved their performance increasing $d$, particularly in validity, but still achieved less than one third the performance of AriEL. VAE is the second best in validity, supporting our hypothesis, that volume coding facilitates retrieval of information by random sampling.

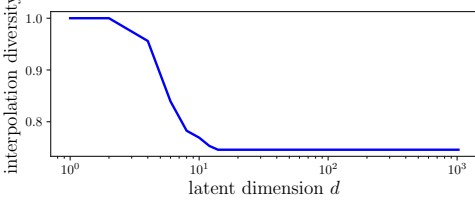

Figure 2: **Interpolations between random points in the latent space of AriEL, and diversity of the sentences generated in between.** For low dimensions all sentences are very densely packed, and in the extreme of one dimension, all sentences are found following one given dimension. As the dimensionality increases, the sentences are redistributed in $[0, 1]^d$ and less sentences are found in a given direction. The lower bound at 0.746 is related to the language complexity.

| | param | prediction accuracy | uniqueness | validity |
|---|---|---|---|---|
| $d = 16$ | | | | |
| AriEL | 2,901K | 92.56% | 57.41% | **29.59%** |
| Transformer | 588K | 87.91% | **96.75%** | 1.96% |
| AE | 2,787K | 11.97% | 13.68% | 2.67% |
| VAE | 2,787K | 13.15% | 6.26% | 1.61% |
| $d = 512$ | | | | |
| AriEL | 2,901K | 92.54% | 55.46% | **22.47%** |
| Transformer | 18,809K | 86.9% | 50.33% | 5.94% |
| AE | 4,900K | 15.45% | 12.4% | 2.68% |
| VAE | 5,425K | 32.51% | **98.51%** | 0.2% |

Table 4: **Performance on the GuessWhat?! Questioner data.** For the real dataset the pattern is repeated: AriEL shows that a larger value of valid sentences is possible
. Transformer 16 gave better results than when $n_{layers}$ was increased from 2 to 20 to increase its learnable parameters from 588K to 2,666K.

## 5  DISCUSSION

**AriEL latent space $d$ is a free parameter.** It is worth to stress that the size $d$ of the latent space of AriEL can be defined at any time, for a fixed Language Model. It could therefore be controlled with a learnable parameter, with the activity of another neuron, or as another function of the input. In fact, as we increase $d$, the volumes will have more neighbouring volumes that represent different sentences, as confirmed by the interpolation study, figure 2. It could have implications as well during training e.g. to have a gradient that can rely more on its angle than on its magnitude.

**What to choose for a learning agent with a language module?** Our study suggests that a learning agent that needs a language model to interact with other agents, would benefit from AriEL to generate a diverse language. It outperformed the rest both in the toy dataset and on the real data.

**Partial evidence for volume codes.** The experiments performed suggest that the volume aspect of AriEL is to be held responsible of its success. We have provided evidence on how volume coding can be beneficial for retrieval of stored information that is composed of discrete symbols, and variable length, by random sampling, in contrast with simply distributed representations. It is in fact AriEL to generate more valid sentences, an explicit volume coding method. VAE is the second on the toy dataset, an implicit volume coding method, but it performed poorly on the real dataset. The low prediction accuracy of AE/VAE, has to be read in conjuction with the grammar accuracy: it basically means that those methods are overfitting the training data, and even if they often manage to produce grammatically correct sentences when a test sentence is given at the input, the volumes/points that would represent new test sentences, seem to have disappeared, all the latent space is dedicated to only the training set.

**Transformers are hard to sample from the latent space.** The Transformer has been used in this work in an uncommon way: by sampling its latent space instead of its input space. Its low validity score reflects that. Our aim was to better understand the latent organization of language, so, we do not want to suggest this is the most effective way to use Transformer. Transformer is excellent when sampled in the input space, but it's difficult to sample from the latent space. This is so because Transformer represents each word by a $d$ dimensional vector while the other approaches represent whole sentences in $d$ dimensional vectors, Transformer needs an extremely high dimensional vector to represent a sentence, $n \cdot d$ where $n$ is the number of words in a sentence. This makes it extremely hard to find sentences using uniform random sampling in the latent space.

## 6 Conclusion and Future Work

We proposed AriEL, a volume mapping of language into a continuous hypercube to be used as a reference system. It provides a latent organization of language that excels at several metrics related to the use of language, and especially at generating many unique and grammatically correct sentences sampling uniformly the latent space. AriEL fuses arithmetic coding and k-d trees to construct volumes that preserve the statistics of a dataset. In this way we construct a latent representation that assigns a data sample to a volume, instead of a point. When compared to standard techniques it highlights room for improvement in their capacity for generation, prediction and generalization.

Recurrent-based continuous sentence embeddings largely overfit the training data and only cover a small subset of the possible language space, particularly when the size of the latent space is small. They also fail to learn the underlying CFG and generalize to unbiased sentences from that CFG. However they manage to generate quite a few diverse valid sentences. Transformer managed to avoid overfitting even after being overtrained, proving its robustness. It performed a remarkable generalization to the unbiased data. However it proves hard to use as a generator from the continuous latent space using random sampling.

On the one hand, this study helps to realize how much of the latent space lies unused by standard architectures. On the other hand, AriEL can be seen as a technique to provide an effective interface between multi-modal RL agents that need a pretrained language model for language interaction. We stress that volume based codes can provide an advantage over point codes in generation tasks. AriEL allows us to sample/generate in theory the same probability distribution as the training set and in practice a more diverse set of sentences, as demonstrated on the toy and on the human dataset.

Our planned next step is to use AriEL as a module in a learning agent. This study has been performed for dialogue based language generation, which implies short sentences. It would be useful for the NLP community to understand if this method generalizes to the compression of longer texts.

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

# SUPPLEMENTARY MATERIAL

## 1 CONTEXT-FREE GRAMMAR (CFG) USED IN THE EXPERIMENTS

The context free grammar used to generate the biased and unbiased sentences is composed by the following rules:

```
s  →  q

q  →  qword adjective ',' adjective 'and' adjective '?'
q  →  qword adjective 'and' adjective '?'
q  →  qword adjective '?'
q  →  qword 'made' 'of' noun_material '?'
q  →  qword preposition np '?'
q  →  qword np '?'

np  →  determiner adjective adjective adjective noun
np  →  determiner adjective ',' adjective 'and' adjective noun
np  →  determiner adjective 'and' adjective noun 'made' 'of' noun_material
np  →  determiner adjective adjective noun
np  →  determiner adjective 'and' adjective noun
np  →  determiner adjective noun 'made' 'of' noun_material
np  →  determiner noun 'made' 'of' noun_material
np  →  determiner adjective noun
np  →  determiner noun

qword  →  'is' 'it'  |  'is' 'the' 'object'  |  'is' 'the' 'thing'
noun  →  noun_object  |  noun_material  |  noun_roomtype
preposition  →  preposition_material

adjective  →  adjective_color  |  adjective_affordance  |  adjective_overall_size  |
              adjective_relative_size  |  adjective_relative_per_dimension_size  |
              adjective_mass  |  adjective_state  |  adjective_other

noun_object  →  'accordion'  |  'acoustic' 'gramophone'  |  'bar'  |  'barrier'  |
                'basket'  |  'outdoor' 'lamp'  |  'outdoor' 'seating'  |  ...

noun_material  →  'bricks'  |  'carpet'  |  'decoration' 'stone'  |  'facing' 'stone'  |
                  'grass'  |  'ground'  |  'laminate'  |  'leather'  |  'wood'  |  ...

noun_roomtype  →  'aeration'  |  'balcony'  |  'bathroom'  |  'bedroom'  |  'boiler' 'room'  |
                  'garage'  |  'guest' 'room'  |  'hall'  |  'hallway'  |  'kitchen'  |  ...

determiner  →  'a'  |  'an'  |  'that'  |  'the'  |  'this'

preposition_material  →  'made' 'of'

adjective_color  →  'antique' 'white'  |  'magenta'  |  'maroon'  |
                    'slate' 'gray'  |  'white'  |  'yellow'  |  ...

adjective_affordance  →  'actable'  |  'addable'  |  'addressable'  |  'deliverable'  |
                         'destroyable'  |  'dividable'  |  'movable'  |  ...

adjective_size  →  adjective_overall_size  |  adjective_relative_per_dimension_size

adjective_overall_size  →  'average-sized'  |  'huge'  |  'large'  |  'small'  |  'tiny'
adjective_relative_per_dimension_size  →  'deep'  |  'narrow'  |  'shallow'  |
                                          'short'  |  'tall'  |  'wide'

adjective_mass  →  'heavy'  |  'light'  |  'moderately' 'heavy'  |  'moderately' 'light'  |
                   'slightly' 'heavy'  |  'very' 'heavy'  |  'very' 'light'
adjective_state  →  'closed'  |  'opened'
adjective_other  →  'textured'  |  'transparent'
```

## 2  Size of the language space

From the CFG used in the experiment, it is possible to extract a total of 15,396 distinct grammar rules, some are shown below. However, for simplicity, we defined only 4, related to the number of adjectives in it. In the case of the unbiased dataset, those rules can produce a total of 9.81e+18 unique sentences. The total number of unique sentences for the biased dataset is expected to be an order of magnitude smaller.

```
[qword, prep_material, determiner, adj_state, 'and', adj_other, noun_roomtype, '?']
[qword, prep_spatial, determiner, adj_other, adj_state, adj_state, noun_object, '?']
[qword, determiner, adj_other, ',', adj_mass, 'and', adj_affordance, noun_roomtype, '?']
[qword, determiner, adj_relative_per_dimension_size, adj_overall_size, noun_object, '?']
[qword, determiner, adj_overall_size, ',', adj_state, 'and', adj_state, noun_material, '?']
[qword, prep_spatial, determiner, adj_other, adj_mass, adj_affordance, noun_material, '?']
[qword, adj_state, 'and', adj_relative_size, '?']
[qword, prep_material, determiner, adj_mass, adj_other, adj_other, noun_material, '?']
[qword, prep_spatial, determiner, adj_state, adj_other, adj_color, noun_object, '?']
[qword, determiner, adj_relative_size, 'and', adj_overall_size, noun_material, '?']
[qword, determiner, adj_state, adj_overall_size, adj_other, noun_roomtype, '?']
[qword, determiner, adj_other, adj_state, adj_mass, noun_material, '?']
[qword, determiner, adj_overall_size, 'and', adj_other, noun_material, '?']
[qword, determiner, adj_color, adj_other, noun_object, '?']
[qword, prep_spatial_rel, determiner, adj_mass, adj_color, noun_roomtype, '?']
[qword, determiner, adj_state, 'and', adj_relative_size, noun_object, '?']
[qword, determiner, adj_color, adj_color, adj_relative_size, noun_material, '?']
[qword, determiner, adj_affordance, noun_object, '?']
[qword, determiner, adj_other, adj_other, adj_state, noun_roomtype, '?']
```

## 3  Example of sentences generated from the CFG

### 3.1  Biased sample sentences

- is it large , light yellow and light ?
- is it white , deep pink and average-sized ?
- is it a light , huge and shallow laminate ?
- is the object average-sized and light ?
- is the object fashionable , ghost white and pale turquoise ?
- is the thing huge , huge and khaki ?
- is the thing small , ignitable and very light ?
- is the object a notable very light orange carpet ?
- is the object this small wood made of facing stone ?
- is the object a textured and combinable floor cover made of laminate ?

### 3.2  Unbiased sample sentences

- is the object the huge tiny lovable guest room ?
- is the object the closed closed transparent textile ?
- is the thing a transparent , narrow and slightly heavy textile ?
- is it steerable , dark orange and light ?
- is it gray , very heavy and textured ?
- is it closed , heavy and moderately light ?
- is it transparent , transformable and moderately light ?
- is the thing average-sized and dark red ?
- is the thing large and deep garage ?
- is it that slightly heavy stucco made of grass ?

# 4 VOCABULARY

| Annotation | Nb. of classes | Example of classes |
|---|---|---|
| Noun | 86 | air conditioner, mirror, window, door, piano |
| WordNet category (Miller, 1995) | 580 | instrument, living thing, furniture, decoration |
| Location | 24 | kitchen, bedroom, bathroom, office, hallway, garage |
| Color | 139 | red, royal blue, dark gray, sea shell |
| Color property | 2 | transparent, textured |
| Material | 15 | wood, textile, leather, carpet, decoration stone |
| Overall mass | 7 | light, moderately light, heavy, very heavy |
| Overall size | 4 | tiny, small, large, huge |
| Category-relative size | 10 | tiny, small, large, huge, short, shallow, narrow, wide |
| State | 2 | opened, closed |
| Acoustical capability | 3 | sound, speech, music |
| Affordance | 100 | attach, bend, divide, play, shake, stretch, wear |

Table S1: Description of vocabulary used.

# 5 USE OF LATENT SPACE

In figure S1, each dot represents a sentence in the latent space. In the first row the dot in the latent space is passed as input to the decoder, while in the second and third row the dot is the output of the encoder when the biased test sentence is fed at its input. Two random axis in $\mathbb{R}^d$ are chosen for the generator, first row, while two axis were chosen subjectively among the first components of a PCA for the encoder, second and third row. In every case, the values in the latent space where normalized between zero and one to ease the visualization. Lines are used to ease the visualization of the clusters and shifts of data with their label, since the point clouds overlap and are hard to see. The curves are constructed as concave hulls of the dots based on their Delaunay triangulation, a method called alpha shapes Edelsbrunner et al. (1983).

We can see in figure S1 (first row) how easy it is to find grammatical sentences when randomly sampling the latent space for each model. AriEL practically only generates grammatical sentences and AE and VAE perform reasonably well too, while Transformer fails. AriEL failures are plot on top, to remark how few they are, while AE and VAE failures are plot at the bottom, otherwise they would hide the rest given how numerous they are. In the same figure (rows two and three) we can observe how different methods structure the input in the latent space, each with prototypical clusters and shifts. The Transformer presents an interesting structure of clusters whose purpose remains unclear. Interestingly, the encoding maps seem to be more organized than the decoding ones. All the models seem to cluster or shift data belonging to different classes at the encoding, that could be taken advantage of by a learning agent placed in the latent space. However it seems hard to use the Transformer as a generator module for an agent. The good performance of AriEL is a consequence of the fact that all the latent space is utilized, and in no directions large gaps can be observed. This can be seen in the two encoding rows, where the white spaces around the cloud of dots are consequence of the rotation performed by the PCA, otherwise all the space between 0 and 1 would be utilized by AriEL.

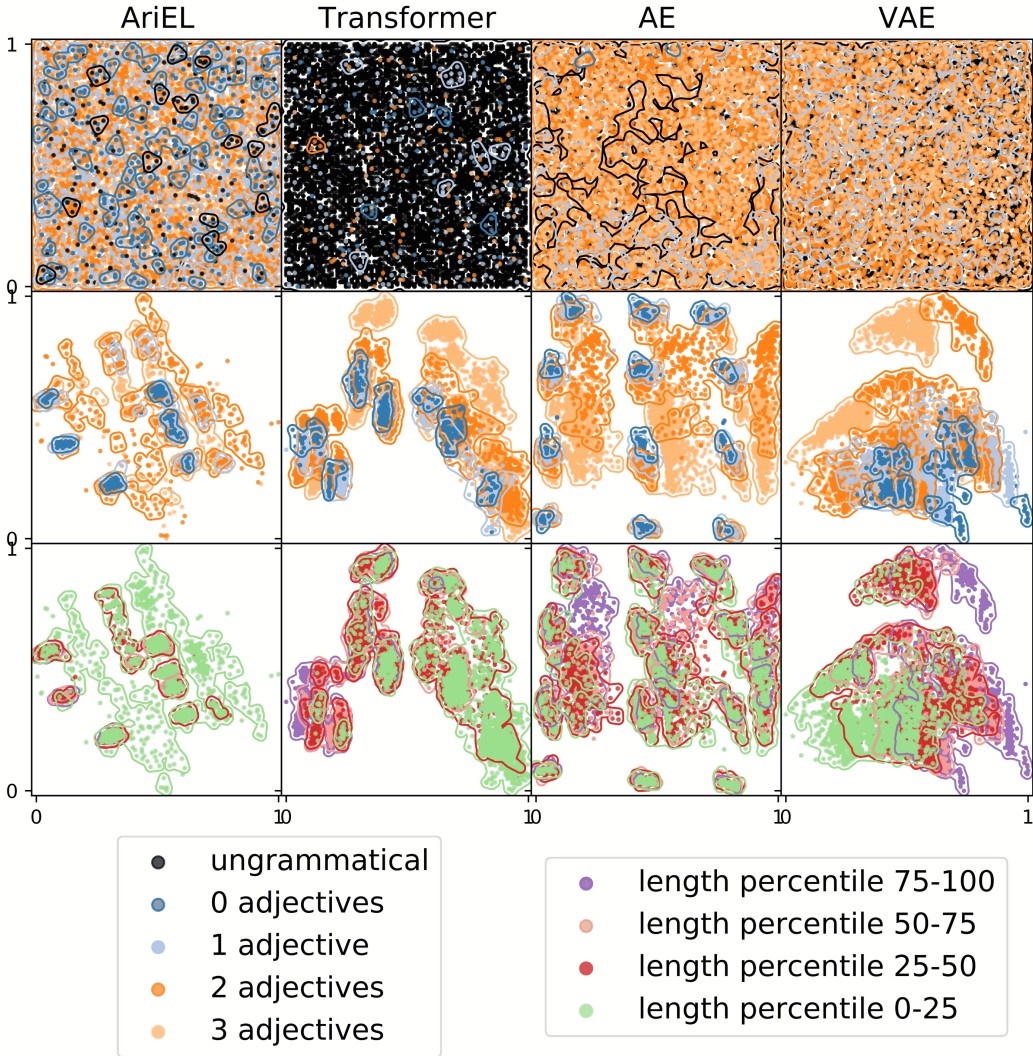

Figure S1: **Random-sampling-based generation in the first row, and encoding of input sentences in the remaining rows.** A sentence is represented by a point in the latent space. First row shows the proportion of grammatically correct sentences that can be decoded by random uniform sampling the latent space. AriEL sampled almost only grammatical sentences (ungrammatical are so few that are placed on top in the plot). Transformer mainly yielded ungrammatical sentences, while AE and VAE were able to produce many grammatical sentences (ungrammatical are below, otherwise they would cover up the grammatical). Each dot is labeled according to how many adjectives the sentence generated has. Second and third rows show the clusters of points in the latent space for the test sentences as they are mapped by the encoders. All models seem to shift the clusters to some degree according to the number of adjectives in the sentence, in the second row. A similar conclusion applies to the third row, that shows where sentences of different length are encoded. For all panels, we searched subjectively for the dimensions that would better reveal some clustering, with the help of PCA. We scaled all latent representations between [0,1] for visualization.

# 6 VISUALIZATION OF PERFORMANCE ON TOY DATA

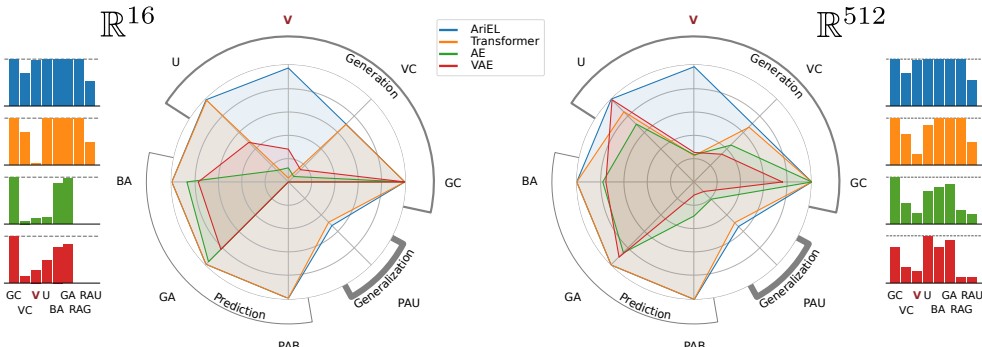

Figure S2: **Radar Chart of the Quantitative Assessment. Latent space of $\mathbb{R}^{16}$ on the left and $\mathbb{R}^{512}$ on the right.** Training was performed on biased sentences. The metrics are defined in Methodology: Generalization is measured by *prediction accuracy of unbiased* sentences (PAU), Prediction by *prediction accuracy of biased* sentences (PAB), *grammar accuracy* (GA) and *bias accuracy* (BA) and Generation by *uniqueness* (U), *validity* (V), *vocabulary coverage* (VC) and *grammar coverage* (GC). AriEL excels in all the 8 metrics. Most importantly AriEL outperforms every other method in Generation Validity (V) and it doesn't require a large latent space to do so ($\mathbb{R}^{16}$ similar to $\mathbb{R}^{512}$). VAE performs remarkably well at generating unique and grammatical sentences (validity, V) when the latent space is small ($\mathbb{R}^{16}$), probably given the volume-code nature of the method. Transformer performs exceptionally at not overfitting in the reconstruction tasks and generalizing, it manages to cover all grammar rules, even with a very small number of parameters ($\mathbb{R}^{16}$). Transformer proved to be an inefficient generator using random sampling as input (validity) but improved with a larger latent space. For a larger latent space of $\mathbb{R}^{512}$, AE and VAE overfit less (PAU and PAB) and improve their Generation (V).

# 7 AriEL algorithm

| **Algorithm 1 AriEL Encoding** | **Algorithm 2 AriEL Decoding** |
|---|---|

**Input:** sentence: $S = (s_j)_{j=1}^n$
**Output:** $\mathbf{z}$ represents $S$ in $[0,1]^d$

```
 1: function AriEL_ENCODE(S)
 2:     d = latent space dimension
 3:     B_low = zeros(d)
 4:     B_up = ones(d)
 5:     n = length(S)

 6:     for i = 0, ··· , n − 1 do
                ▷ choose dimension to split
 7:         d_i = i mod d
 8:         P_next(s) = P_LM(s|(s_j)_{j<i})
 9:         c_low(s) = Σ_{s>s'} P_next(s')
10:         c_up(s) = Σ_{s>s'−1} P_next(s')
11:         range = B_up(d_i) − B_low(d_i)
                ▷ update volume bounds
12:         B_up(d_i) = B_low(d_i) + range · c_up(s_i)
13:         B_low(d_i) = B_low(d_i) + range · c_low(s_i)
14:     end for
                ▷ represent the volume by its center
15:     z = (B_low + B_up)/2
16:     return z
17: end function
```

**Input:** $\mathbf{z}$ represents $S$ in $[0,1]^d$
**Output:** sentence: $S = (s_j)_{j=1}^n$

```
 1: function AriEL_DECODE(z)
 2:     d = dimension(z)
 3:     B_low = zeros(d)
 4:     B_up = ones(d)

 5:     S = ⟨START⟩
 6:     for i = 0, ··· , n_max − 1 do
                ▷ choose dimension to unsplit
 7:         d_i = i mod d
 8:         P_next(s) = P_LM(s|S)
 9:         c_low(s) = Σ_{s>s'} P_next(s')
10:         c_up(s) = Σ_{s>s'−1} P_next(s')
11:         range = B_up(d_i) − B_low(d_i)
                ▷ update volume bounds
12:         Bs_up(s) = B_up(d_i) + range · c_up(s)
13:         Bs_low(s) = B_low(d_i) + range · c_low(s)
            ▷ any point in the volume is assigned the
        symbol s_i
14:         s_i = find_s( Bs_low(s) < z(d_i) < Bs_up(s) )
15:         B_up(d_i) = Bs_up(s_i)
16:         B_low(d_i) = Bs_low(s_i)
17:         S = S.append(s_i)
18:     end for
19:     return S
20: end function
```

Figure S3: **Algorithms for AriEL encoder and decoder** B stands for bound, and $B_{up}$ and $B_{low}$ for the upper and lower bounds that define the AriEL volumes, the blue color identifies the lines with the major differences between encoder and decoder and $P_{LM}$ identifies the Language Model inside AriEL. Its cumulative distributions ($c_{up}$, $c_{low}$) are used to define the limits of the volumes and its size ($range$). (Left) AriEL Encoding: from sentence to continuous space. Finally the volumes are represented by their central point $\mathbf{z}$ for simplicity. (Right) AriEL decoding: from continuous space to sentence. $\mathbf{z}$ is used to identify which volume has to be picked next.

# 8 Training details

We go through the training data 10 times, in mini-batches of 256 sentences. We applied teacher forcing (Williams and Zipser, 1989) during training. We use the Adam (Kingma and Ba, 2015) optimizer with a learning rate of 1e-3 and gradient clipping at 0.5 magnitude. Learning rate was reduced by a factor of 0.2 if the loss function didn't decrease within 5 epochs, with a minimum learning rate of 1e-5. For all RNN-based embeddings, kernel weights used the Xavier uniform initialization (Glorot and Bengio, 2010), while recurrent weights used random orthogonal matrix initialization (Saxe et al., 2014). Biases are initialized to zero. Embeddings layers are initialized with a uniform distribution between [-1, 1]. For Transformer the multihead attention matrices and the feedforward module matrices, used the Xavier uniform initialization (Glorot and Bengio, 2010), the beta of the layer normalization uses zeros, and its gamma uses ones for initialization. AE and VAE are trained with a word dropout of 0.25 at the input, and VAE is trained with KL loss annealing

that moves the weight of the KL loss from zero to one during the 7th epoch, similarly to the original work (Bowman et al., 2016).

