# OpenReview forum: "AriEL: Volume Coding for Sentence Generation Comparisons"
_ICLR.cc/2021/Conference — Reject_

### Official Review · AnonReviewer1 · 2020-10-28
**Idea is interesting but evaluation is insufficient**

**Rating:** 3
**Confidence:** 4

**Review:**

This paper proposes a sentence embedding called AriEL. Specifically, based on arithmetic coding and k-d trees, AriEL maps sequences of discrete data into volumes in the latent space, and can then retrieve sequences by random sampling. AriEL is compared to other standard techniques such as Transformer and Variational Autoencoders. Results show that it can generate more diverse and valid sentences.

**Pros:**

The idea of constructing sentence encoders by arithmetic coding is interesting and novel. AriEL takes the frequency of sentences into account, and the frequency implicitly models the statistics of human language. I have not seen prior work in this direction, and it is worth exploring.

**Cons:**
1. My major concern is the evaluation. The evaluation metrics in the paper are very limited and not convincing. They fail to assess the strengths of the proposed sentence embedding. Specific downstream applications are missing here. A solid evaluation should at least include either classification tasks, such as sentiment analysis, textual entailment; or generation tasks, such as simplification, machine translation.


2. The paper is hard to follow and the writing has greatly impeded understanding. I will mention a few that confuse me most:\
(1) In the second paragraph of Section 3.2, what does “but the probabilities defined by the latter are used as a deterministic Russian roulette” mean? Does that mean the argmax is one-hot?\
(2) What is the purpose of defining biased and unbiased sentences? And in Table 1, what does “comply with the bias” mean?\
(3) In Section 3.4.2, I can hardly believe the definition of “grammar coverage”. Why is the number of adjective able to represent grammar rules?


3. I doubt only modeling the frequency of sentences is enough for a sentence encoder. There are many aspects of language (e.g. taxonomy, lexicon) that should be considered. But again, these are to be proved by downstream tasks.

Overall, I suggest rejecting the paper in its current state.
***
UPDATE: The authors have addressed some of my comments. I appreciate their efforts for making the paper clearer. That being said, I would still keep my original score, because my major concern (evaluation) has not been addressed. Also, the paper would be much better if its writing and organization can be improved.

---

> ### Author Response · Authors · 2020-11-17
> **first reply**
>
> 1. reply to paragraph 'The idea of constructing...'. Thanks!
> 2. reply to paragraph 'My major concern...'. That's fair. We are trying to either import GPT2 as a language model inside AriEL, using it in a dialogue game, or train a language model on WikiText.
> 3. reply to paragraph 'In the second paragraph...'. You are right, we have to clarify that statement. I don't know yet what's going to be the best way to phrase it, but we will think about it.
> 4. reply to paragraph 'What is the purpose...'. The purpose is to understand what is happening in latent space, if only the training set is represented, if only sentences in the same distribution of the training set are represented (biased), or if we can go a bit out of distribution, and see if grammatically correct but unbiased sentences, can still be represented by the network. But it's true that it should be commented in more detail in the discussion. I think framing it in terms of out/in distribution, is more common, so we should explain it like that. What we mean by 'comply with the bias' is that the sentence belongs to the CFG, and the words in the sentence can be together according to the adjacency matrix for words generated for the experiments. The bias is defined in section 3.3, second paragraph of the toy dataset. In table 1, the input sentences are unbiased and when the network tries to reconstruct them, it might fail: the failures can be biased or unbiased. But we should add a line in section 3.3 to let the reader understand at a practical level, how one uses the adjacency matrix to evaluate if a sentence has the correct bias or not.
> 5. reply to paragraph 'In Section 3.4.2, I can hardly believe...'. Since the context-free grammar is a rather complex tree of possible outcomes, with placeholders and terminal symbols, there's a high degree of ambiguity on what constitutes one grammar rule. For example if you have a look at the grammar in section 1 from the Supplementary Material, we could have e.g. written all the terminal nouns after the placeholder 'noun', but we chose to split that placeholder in the three placeholders 'noun object', 'noun material' and 'noun room type'. If we specify how many branches the grammar tree has, it has 15K, as mentioned in section 2 from the Supplementary Material. We thought however that it would make it simpler to explain and more efficient computationally to check, if we simplified our definition of grammar rule. That's why, as you said, our definition of grammar rule is rather simple.
> 6. reply to paragraph 'I doubt only modeling the frequency...'. That is true. We focused mainly on sentence generation because it was in line with our research to develop dialogue agents. As we mentioned above, we are trying to have some results for AriEL in a downstream task for the next upload.
>
> Thanks for the review!

---

> > ### Author Response · Authors · 2020-11-24
> > **last rebuttal reply**
> >
> > 1.
> > 2. We have been actively working on having AriEL in a dialogue agent, but we could not make it on time for this rebuttal.
> > 3. The phrasing in the second paragraph of section 3.2 has been changed, to make it more clear.
> > 4. We tried to give a more practical explanation in section 3.3. We changed the phrasing in table 1, into a hopefully more clear explanation.
> > 5. We commented on the pros and cons of our simple definition of a grammar rule in the paragraph before the last one in page 5.
> > 6. As in reply 2.
> >
> > Thanks for your time and insights!

---

### Official Review · AnonReviewer2 · 2020-10-28
**Interesting idea, but not practical for other applications**

**Rating:** 4
**Confidence:** 4

**Review:**

This paper proposes AriEL, a sentence encoding method onto the compact space [0, 1]^d. It leverages essences of arithmetic coding and kd-tree to encode/decode sentences with a fixed region of the space. With the property of arithmetic coding, in theory, it can map sentences with any lengths into individual values, and any points on [0, 1]^d can map back into corresponding sentence. Although the method relies on neural network based LMs to assign sentences into corresponding regions, the generality of mapping between any sentences/points is kept while changing the LM's behavior. The idea is interesting.

However, there is some disadvantages of the proposed encoding which are not always mentioned in the paper. First, due to the topological difference between the proposed encoding and other spaces (e.g., Euclidian space), the proposed encoding could not be treated as embeddings in some usual meanings, e.g., it is hard to calculate the "similarity" between two encodings by arithmetics on real numbers as many deep learning methods implicitly does. Actually, there seems no evidence of advantages of the proposed encodings on other tasks which are not designed for this encoding unlike experiments on the paper.

Second, the resulting encodings will be affected directly by the capability of the actual representation of real numbers. E.g., if we used float32 for each dimension, the [0, 1] space can contain only information up to 30 bits long in the most efficient case, which may be insufficient to encode "all" sentences into the compact space. It will be problematic when we encode very long sentences (|sentence| >> d). Experiments in the paper did not figure out this point enough because the mean number of words in the corpus is too small (9.9 and 5.9 words, whereas d = 16).

There are also several presentation errors in the paper:

- Using different format of all citations.
- Table 3 clearly exceeds the width limit.

---

> ### Author Response · Authors · 2020-11-17
> **first reply**
>
> 1. reply to paragraph 'This paper proposes AriEL...'. Thanks!
> 2. reply to paragraph 'However, there is some...'. On purpose we relax the need of having an euclidean space that uses similarity to place sentences close by. We do it knowing there will be something to lose and something to win. We wanted to focus as well on generation, so decoding, while we hipothesize similarity would be more relevant for embedding as you say, so, encoding. Preliminary explorations seem to suggest that AriEL is well behaved to do some algebra in the latent space, but we didn't explore thoroughly this impression.
> 3. reply to paragraph 'Second, the resulting...'. That is right, we focus on sentence generation, keeping in mind dialogue systems as a goal, but it would be very interesting to see how it applies to longer sentences, and we will mention it earlier in our next upload, instead of mentioning it only in the last paragraph of the conclusion. We are trying to either import GPT2 as a language model inside AriEL, using it in a dialogue game, or train a language model on WikiText. Thanks for the review!

---

> > ### Author Response · Authors · 2020-11-24
> > **last rebuttal reply**
> >
> > 1.
> > 2. We are actively working on having AriEL in a dialogue agent, but we could not make it on time for this rebuttal.
> > 3. Same as the previous point. We added a clarification sentence at the end of the second paragraph in the introduction to position our work more clearly in the dialogue literature.
> > 4. We corrected the citation style.
> > 5. With the time constraint we couldn't make it on time to reformat table 3 and we thought it was better to give all the experimental data we had. In case the article is accepted we will fix it for the final version.
> >
> > Thanks for your time and insights!

---

### Official Review · AnonReviewer4 · 2020-10-29
**interesting paper, would benefit from more rigorous empirical evaluations**

**Rating:** 5
**Confidence:** 3

**Review:**

Summary: This paper proposes Arithmetic coding and k-d trEes for Language (AriEL), a representation learning method for more efficiently mapping discrete data (text) into a continuous space. Specifically, AriEL uses a language model to split the latent space into volumes.

Clarity: The paper is overall easy to follow.

Originality: The idea to incorporate more structure into the learned encodings via arithmetic coding and K-d trees is an interesting one.

Pros: AriEL seems to perform quite well on the toy dataset and the GuessWhat?! dataset.

Cons:
- I think the paper would be more significant if the authors would have demonstrated the efficacy of their method on more benchmark datasets; for example, WMT (Kaiser et al. 2018) or WikiText (Merity et. al 2016).
- I am curious as to how the performance of some of these models are *so* bad: for example, the Transformer obtains a validity score of 4.7% on the toy dataset, and the AE/VAE get roughly 0% accuracy on the prediction task. I would appreciate it if the authors could elaborate more on why exactly AriEL seems to be doing so well, as well as some of its potential limitations. (at least on the experiments side, there doesn't seem to be any, which seems too good to be true?)

Questions:
- Can I understand AriEL to have an encoding/decoding complexity of O(nD^2) rather than the O(n) as written later in the paragraph on page 3, since in practice AriEL will most likely always use a language model? Or is this specific to an RNN? I am wondering if this would hinder the use of AriEL on more complex datasets that require larger hidden sizes (D).

Minor typos:
- "followed by a 140-unit LSTM" (Section 3.2)
- "Then we turn to" (Section 3.1)

---
UPDATE: Thanks to the authors for responding to my questions and updating the submission. I will keep my score as is, since I think that the paper would greatly benefit from more practical/rigorous empirical evaluations to demonstrate the usefulness of the approach.

---

> ### Author Response · Authors · 2020-11-17
> **first reply**
>
> 1. Thanks for the comment on clarity and originality!
> 2. reply to paragraph 'I think the paper would...'. That is true. We do not see how to use it effectively for language translation, since it was developped for language generation giving partially up on the similarity in latent space, that is very important for translation, but we are working on how to plug in GPT2, or using WikiText, as you suggested.
> 3. reply to paragraph 'I am curious as to how...'. True. Some remarks were removed to make the content fit within the 8 pages, but now with the extra page we will describe in more detail the interesting results. About the 0% accuracy of AE/VAE, it has to be read in conjuction with the grammar accuracy: it basically means that those methods are overfitting the training data, and even if they often manage to produce grammatically correct sentences when a test sentence is given at the input, the volumes/points that would represent new test sentences, seem to have disappeared, all the latent space is dedicated to only the training set. About the 4.7% accuracy for the Transformer, given that it represents each word by a d dimensional vector while the other approaches were representing whole sentences in d dimensional vectors, Transformer needs an extremely high dimensional vector to represent a sentence, N*d where N is the number of words in a sentence. This makes it extremely hard to find sentences using uniform random sampling. The simplest way to put it is that Transformer is excellent when sampled in the input space, but it's difficult to sample from the latent space. We don't claim Transformers should be used that way, it's only an experimental finding that helps us understand the latent organization, but I should probably stress this point in the discussion.
> 4. reply to paragraph 'Can I understand AriEL...'. Ok, I think you are right, what we wanted to say instead of time complexity is 'it has a minimum number of sequential operations of O(n)', so we will change it in the next upload. Thanks! And thanks for the review!

---

> > ### Author Response · Authors · 2020-11-24
> > **last rebuttal reply**
> >
> > 1.
> > 2. We are actively working on having AriEL in a dialogue agent, but we could not make it on time for this rebuttal.
> > 3. We added those remarks in the last two paragraphs of the Discussion section.
> > 4. We made the suggested modification in the last paragraph of section 3.1.
> >
> > Thanks for your time and insights!

---

### Official Review · AnonReviewer3 · 2020-10-30

**Rating:** 7
**Confidence:** 4

**Review:**

This paper proposes "volume encoding" for sequence modeling. Unlike traditional autoencoder or variation autoencoder model family, the proposed model AriEL applies KDTree to map the input sequence to a quantized multi-dimensional space as the code, and supports tasks such as reconstruction and generation.

**high level comment**
Indeed this paper proposes a novel and interesting way to do sequence modeling. From its root, data like language are naturally discrete. The traditional methods ubiquitously adopt a continuous code as the representation that poses a gap between discrete and continuous mode. AriEL, on the other hand, represents this type of data into the discrete mode.

From a different perspective, the continuous code does have its own merits. The smoothness of the manifold may allow us to perform vector arithmetic and have better performance at practical tasks like grammar correction or semi-supervised learning. The paper does not include tasks like these. Could the authors clarify?

**table 1**
Can the authors clarify on how to quantitatively or qualitatively measure the correlation between these generations and the training dataset bias?

**table 2**
It's pretty well-known that VAE type of reconstruction may have grammar errors like repeated words. The AriEL model seems to be able to avoid such error but at a cost of wrong n-grams like "small large". Could the authors give some intuition here?

**open discussions**
The current version of AriEL seems to rely on a sequential modeling workhouse like the RNNs. Is it amenable to other types of models?

Would the usage of KDTrees and dataset stats amplifies the training set bias so it might generalize poorer than the rivalling models?

What kind of practical applications can we dream up the AriEL for, e.g. leveraging the volume code?

**conclusion**
With all the questions above, I still vote an accept for the proposed model for its novelty. In my honest opinion, this paper is valuable for the community.

---

> ### Author Response · Authors · 2020-11-17
> **first reply**
>
> 1. reply to paragraph 'From a different perspective...'. Yes, the main focus is on generation, and we purposefully gave partially up on the smoothness of representation to increase the availability of sentences. In some experiments that we performed, KD-trees seemed to be quite well behaved for latent space algebra, but we didn't go deeper in that investigation. We don't want to claim continuous code should not exist anymore, we claim that there might be something to gain in specific tasks giving up on some of their traits. I will be more clear in the discussion and in the introduction.
> 2. reply to paragraph 'Table 1 Can the authors clarify...'. I understand that what you mean is that the definition of bias in section 3.3 should be explained in a more practical level, so we are going to add a sentence there.
> 3. reply to paragraph 'Table 2 It's pretty well-known that...'. It is true that that n-gram is wrong in natural language, but it is correct in the definition of our context-free grammar.
> 4. reply to paragraph 'Open discussions...'. Yeah, AriEL only requires a language model, so GPT3 could be placed inside, or any other language model, and AriEL would derive its time complexity from the underlying language model.
> 5. reply to paragraph 'Would the usage of KDTrees...'. We wanted to propose a method to evaluate how much of the latent space is used by standard architectures, but we do envision it as potentially interesting for generation by small networks. Since it provides an interface in continuous space with a language model, we developped it to be able to use it to integrate pretrained language models into multi-modal RL agents in our future research, and for those that might find it useful.
> 6. reply to paragraph 'What kind of practical...'. On the one hand, it helps to realize how much of the latent space lies unused by standard architectures. On the other hand, we see it as a technique to provide an effective interface between multi-modal RL agents that need a pretrained language model for language interaction.
> 7. reply to paragraph 'conclusion...'. Great! Thanks a lot! And thanks for the review!

---

> > ### Author Response · Authors · 2020-11-24
> > **last rebuttal reply**
> >
> > 1. We are actively working on having AriEL in a dialogue agent, but we could not make it on time for this rebuttal.
> > 2. We added a sentence in the description of the bias in the toy dataset, to give a more practical explanation on how the bias is created.
> > 3.
> > 4.
> > 5. We are working on having AriEL in a dialogue agent, but we couldn't have it on time for this rebuttal.
> > 6. We added that justification in the conclusion, the paragraph before the last one.
> > 7.
> >
> > Thanks for your time and insights!

---

### Official Review · AnonReviewer5 · 2020-11-07
**Well written paper, but the motivation and evaluation are not clear**

**Rating:** 6
**Confidence:** 2

**Review:**

This paper describes a novel methodology of volume coding for encoding and decoding sentences. The algorithm is based on arithmetic coding.

In general, I believe this is a well written paper. However, I couldn't really understand the content until I read the encoding and decoding algorithms, which are in Section 7 of supplemental material. I strongly think the structure of this paper needs revision, so that people without background of arithmetic coding can understand the core method.

Other than the writing, I still have concerns about the motivation and evaluation. In the first point of contributions, the author states that the proposed method improves the retrieval of learned pattern with random sampling. Does this mean when the coding is randomly sampled, we can see more valid sentences comparing to other methods? If so, the validity in evaluation is the key to claim this contribution.

In Table 3,  the validity percentage of a Transformer with 512 latent dimensionality is only 17.2%. This low score strongly contrasts with our knowledge that a well-trained Transformer language model is very strong at producing valid sentences. One hypothesis is that the amount of training data is not sufficient for Transformer. If this is the case, the proposed method may lose its edge when the training data is abundantly available. Upon reading the experiment section, I couldn't find an explanation.

When comparing with VAE, it's important to compare the interpretability of the latent variables, which is the main purpose we train a generative model. However, if the interpretability is not a major motivation of volume coding, then I'm concerning whether it's meaningful to compare with VAE.

Considering all these factors, I decide to give a weak acceptance to this paper.

---

> ### Author Response · Authors · 2020-11-17
> **first reply**
>
> 1. reply to paragraph 'In general I believe...'. Since now we have one extra page to fill, we could move the algorithm from the supplementary material to page 3 if you think it would improve clarity.
> 2. reply to paragraph 'Other than the writing...'. Yes, exactly! the validity in evaluation is the key to claim this contribution.
> 3. reply to paragraph 'In Table 3...'. Probably the simplest way to put it is that Transformer is excellent when sampled in the input space, but it's difficult to sample from the latent space. This is so because Transformer represents each word by a d dimensional vector while the other approaches represent whole sentences in d dimensional vectors, Transformer needs an extremely high dimensional vector to represent a sentence, N*d where N is the number of words in a sentence. This makes it extremely hard to find sentences using uniform random sampling. We don't claim Transformers should be used that way, it's an experimental finding to clarify how the latent space is used, so I will stress this point in the discussion.
> 4. reply to paragraph 'When comparing with VAE...'. Interpretability and semantic similarity are properties that are purposefully not used in AriEL latent space, as you said, but VAE seems still to have an interesting way to fill the latent space. We conjecture that this is due to its volumetric nature, which seems to be confirmed on toy data but confuted on the human data.
>
> Thanks for the review!

---

> > ### Author Response · Authors · 2020-11-24
> > **last rebuttal reply**
> >
> > 1. Finally with the new additions we didn't have space to add the algorithm to the main article, so we kept it in the Supplementary Materials.
> > 2.
> > 3. We added this clarification in the Discussion section, last paragraph.
> > 4.
> >
> > Thanks for your time and insights!

---

### Decision · Program_Chairs · 2021-01-07
**Final Decision**

**Decision:**

Reject

**Comment:**

the authors propose to use volume coding to enable uniform sampling from an implicit latent space to be used together with a autoregressive language model. all the reviewers find this approach interesting, but all found that the submission would be much stronger with more thorough evaluation. in particular, i noticed that the reviewers wanted to see how the proposed ariel works in comparison to e.g. VAE on a more diverse set of benchmarks, since the choice of two datasets, one synthetic and one small, narrow-domain, is somewhat limited largely due to their relative simplicity. furthermore, the reviewers were unsure whether various evaluation metrics the authors have used are exhaustive nor appropriate to demonstrate the efficacy of Ariel or to put the proposed approach correctly in the context of other approaches. i agree with the reviewers on both of these points.

i'm thus recommending this manuscript be rejected, and strongly recommend the authors give a bit more thoughts on how to demonstrate the effectiveness of the proposed approach in the context of other approaches and the problem of sentence generation (which is the main problem the authors claim to tackle, as the title directly suggests.) with a better planned experiment and analysis, i believe the authors' efforts will have significant impact.